# Deep Learning for Micro-Scale Crack Detection on Imbalanced Datasets Using Key Point Localization

## Abstract

Internal crack detection has been a subject of focus in structural health monitoring. By focusing on crack detection in structural datasets, it is demonstrated that deep learning (DL) methods can effectively analyse seismic wave fields interacting with micro-scale cracks, which are beyond the resolution of conventional visual inspection. This work explores a novel application of DL based key point detection technique, where cracks are localized by predicting the coordinates of four key points that define a bounding region of the crack. The study not only opens new research directions for non-visual applications but also effectively mitigates the impact of imbalanced data which poses a challenge for previous DL models, as it can be biased toward predicting the majority class (non-crack regions). Popular DL techniques, such as the Inception blocks are used and investigated. The model shows an overall reduction in loss when applied to micro-scale crack detection and is reflected in the lower average deviation between the location of actual and predicted cracks, with an average IOU being 0.511 for all micro cracks (>0.00 μm meters) and 0.631 for larger micro cracks (>4 μm).

## 1 Introduction

Structural health monitoring is a critical area where the safety of infrastructures depends on the detection of internal cracks, which are not visible on the surface. These hidden cracks pose a significant challenge because detecting them requires a combination of domain expertise in both deep learning and material science to effectively extract the relevant features (Zhou et al., 2023; Liu & Zhang, 2019). Moreover, processing the numerical data generated demands high computational power, as the task involves analyzing wave propagation through materials and identifying changes caused by cracks. The ability to automate and enhance this detection process is vital for ensuring structural safety in industries such as civil engineering, aerospace, and manufacturing (Cha et al., 2024; Chen & Jahanshahi, 2018). In recent years, deep neural networks have significantly advanced the field of computer vision, particularly in areas like object detection(Krizhevsky et al., 2017). Traditionally, these networks were designed to grow deeper by adding more layers to capture increasingly complex patterns in data. However, this approach faces challenges such as vanishing gradients, increased computational costs, and difficulties in training. As a result, researchers began exploring the idea of wide networks, which emphasize increasing the number of neurons in each layer rather than stacking more layers. This shift has proven especially beneficial for tasks like object detection, where wider networks can capture more detailed features across a broader spatial rangeZagoruyko & Komodakis (2017), improving performance without the complications that come with deeper architectures (He et al., 2015). In this paper, we explore the transition from deep to wide convolutional networks in the context of object detection, specifically for crack detection using numerical data. Wide networks, with their ability to capture diverse features, offer advantages in handling spatial correlations across larger regions of the input, leading to more accurate detection results (Huang et al., 2018; Xie et al., 2017). Unlike traditional deep learning models in computer vision, we successfully detected patterns in numerical wave propagation data, demonstrating that our model can effectively identify and locate cracks. To the best of our knowledge, no prior work has applied this approach to crack detection, making this the first study to use an object detection model specifically for crack detection from numerical data. This work is a significant first step, proving that the model

can detect cracks and accurately determine their location. However, due to the limited availability of data, we were unable to evaluate the model against samples with multiple or more complex cracks. In future work (Section 3.11), we aim to test the model with samples containing multiple cracks and more intricate crack patterns. This paper opens the door to further research on improving crack detection using wide convolutional networks and addressing more complex scenarios in structural health monitoring.

## 2 RELATED WORK

In recent years, deep learning models, particularly convolutional neural networks (CNNs), have emerged as powerful tools for accurately segmenting crack regions from input data (He et al., 2015; Zhang et al., 2016). CNNs have been successfully applied to surface crack detection by learning hierarchical features, enabling improved performance in complex environments. However, segmentation-based methods often face challenges, particularly class imbalance Ghosh et al. (2024); Saini & Susan (2023), where cracks occupy a small number of pixels relative to the background, making it inefficient to train models effectively. This imbalance often leads to poor detection performance as models tend to become biased towards background pixels. Various strategies have been introduced to address class imbalance, such as weighted loss functions, data augmentation, and focal loss (Lin et al., 2018). While these techniques offer some improvements, researchers have shifted attention toward object detection models, including Faster R-CNN, YOLO, and SSD, which have been widely applied in object detection tasks (Redmon et al., 2016; Xie et al., 2021). For example, Yadav et al. Gupta et al. (2023) demonstrated that object detection models like SSD MobileNet require significantly less computational power and are less complex than segmentation models, making them ideal for scenarios where speed and computational resources are limited. This makes object detection an attractive approach for crack detection using numerical data, especially in environments with constrained resources. Object detection models focus on identifying regions of interest, such as cracks, rather than classifying every pixel in the image, which makes them more efficient in handling small objects like cracks (Zhao et al., 2019; Amjoud & Amrouch, 2023; Sun et al., 2024). However, most of these approaches have been applied to visual data, leaving a gap in adapting these techniques to numerical data, such as wave propagation simulations, for detecting internal cracks. In addition to object detection, some researchers have explored the use of numerical data for damage detection in composite materials. For instance, Azuara et al. Azuara et al. (2020) employed a geometric modification of the RAPID algorithm, leveraging signal acquisition between transducers to accurately localize and characterize damage. This highlights the potential of using numerical data in crack detection, further motivating the need to explore object detection models in this context.

## 3 METHOD

### 3.0.1 WIDE CONVOLUTIONAL NETOWORKS

Wide networks, like those incorporating Inception modules Szegedy et al. (2014), provide an alternative to the deeper convolutional networks seen in architectures like ResNetHe et al. (2015) and DenseNet. While deeper networks are highly effective at feature extraction, they come with the trade-off of requiring significantly higher computational power and memory. These deep models extract hierarchical features through successive layers, capturing intricate patterns, but their depth can lead to increased training complexity and higher resource demands. This is where wide networks come into play. Wide convolutional networks address this issue by using multiple convolution layers with different filter sizes within the same layer, enabling efficient and diverse feature extraction without the need for extreme depth. Instead of stacking many layers to extract complex features, wide networks process input data at multiple scales simultaneously. This approach allows for capturing both fine-grained (intrinsic) details and larger, more abstract patterns within the same stage, similar to deep networks, but with improved computational efficiency. Thus, wide networks offer a balance between the high feature extraction capabilities of deep networks like ResNetHe et al. (2015) and DenseNet Moreh et al. (2024), and the need for computational efficiency.

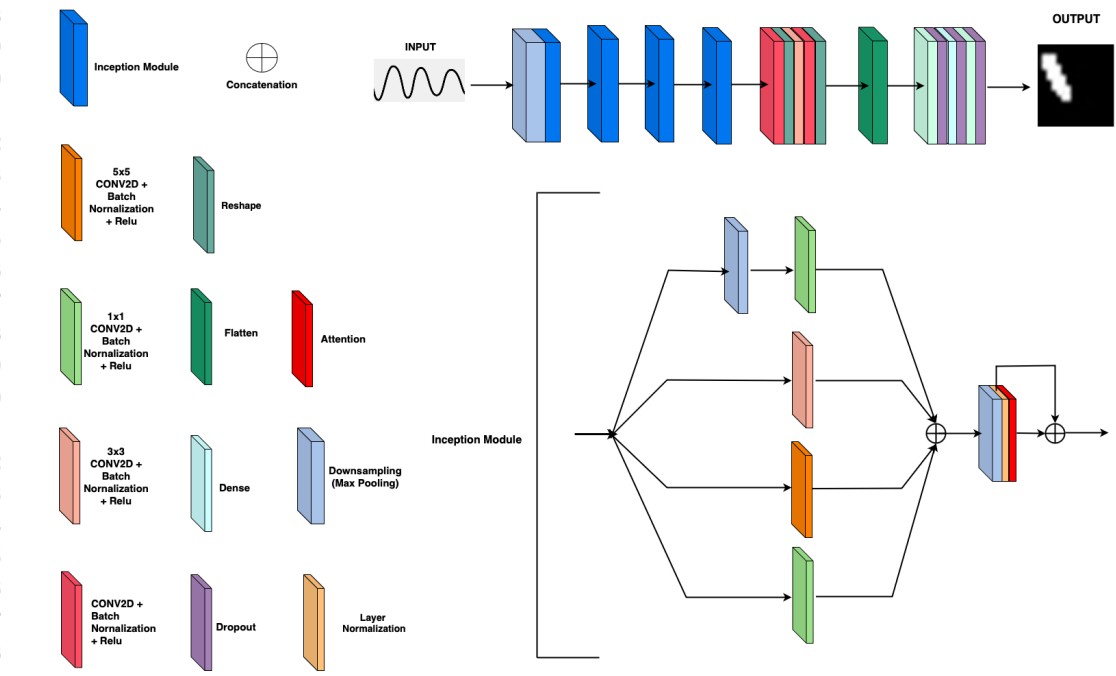

Figure 1: Proposed model architecture for key point localization

## 3.1 MODEL ARCHITECTURE

Our model follows the design philosophy of a wide convolutional networks, The model in this research is designed around a combination of convolutional blocks, attention mechanisms, and dense layers to effectively process and extract features from the input. Each convolutional block is designed with three convolutional branches—1x1, 3x3, and 5x5 convolutions—each processing the input at different scales. These branches are followed by batch normalization and an ReLU activation to stabilize and enhance learning. A max-pooling branch is also included to further downsample the input. The outputs of all branches are concatenated to form a comprehensive feature map, capturing multi-scale information. The attention layer processes the output of the pooling layers by focusing on key areas within the feature map. It calculates an attention-weighted sum of the features, which is then added back to the original input, reinforcing relevant features and filtering out noise. After multiple attention and pooling layers, the feature map is reshaped and processed through two additional convolutional layers. The first uses a kernel size of (31, 1), followed by reshaping and applying a 3x3 and 4x4 convolution. These layers aim to extract more complex patterns from the feature map while further reducing its dimensionality. The fully connected layers consist of dense layers with dropout regularization to prevent overfitting. The first two dense layers have 128 and 64 neurons, respectively, with ReLU activation. The final output layer contains four neurons and uses a linear activation function to provide the regression outputs. The model outputs a four-dimensional vector, representing the final predicted values for the task. The model is optimized using regularization techniques to improve generalization.

## 3.2 LOSS FUNCTIONS

- MSE Loss: The Mean Squared Error (MSE) is a widely used loss function in regression tasksKato & Hotta (2021), measuring the average squared difference between predicted and actual values. The MSE equation is:

$$\text{MSE} = \frac{1}{N} \sum_{i=1}^{N} (y_i - \hat{y}_i)^2$$

where $y_i$ represents the actual values and $\hat{y}_i$ are the predicted values. One of the main advantages of MSE is its strong emphasis on larger errors due to the squaring term, mak-

ing it more sensitive to outliers compared to Mean Absolute Error (MAE). This property ensures that significant deviations have a proportionally higher influence on the loss value, prompting the model to prioritize reducing large errors.

- MAE Loss: The Mean Absolute Error (MAE) is a loss function that calculates the average magnitude of the absolute differences between predicted and actual values. Its mathematical formulation is:

$$\text{MAE} = \frac{1}{N} \sum_{i=1}^{N} |y_i - \hat{y}_i|$$

where $y_i$ represents the actual values and $\hat{y}_i$ are the predicted values. MAE is advantageous over Mean Squared Error (MSE) in several ways. First, MAE is less sensitive to outliers since it does not square the errors, making it more robust in scenarios where large deviations exist. In contrast, MSE, due to its sensitivity to large errors, may cause computational issues in deep neural networks and can be less effective in handling noisy data.

- Huber Loss: The Huber loss combines the advantages of both Mean Absolute Error (MAE) and Mean Squared Error (MSE). Its key feature is its ability to behave like MSE for small errors (thereby benefiting from strong convexity and fast convergence) and like MAE for larger errors (providing robustness against outliers). The formula for Huber loss is:

$$L_\delta(x) = \begin{cases} \frac{1}{2}x^2 & \text{for } |x| \leq \delta \\ \delta(|x| - \frac{1}{2}\delta) & \text{for } |x| > \delta \end{cases}$$

Here, $\delta$ is a threshold that determines the point where the loss function transitions from quadratic to linear behavior. For errors smaller than $\delta$, Huber loss acts like MSE, focusing on fast convergence and sensitivity to minor deviations. For larger errors, it switches to a linear form similar to MAE, preventing outliers from disproportionately affecting the model.

## 3.3 TRAINING PROCEDURE

The training procedure of the model begins with a preprocessing step that utilizes a 1D MaxPooling layer to reduce the temporal dimension of the input, which is originally shaped (2000, 81, 2), by a factor of 4, resulting in a feature map of shape (500, 81, 2). This strong reduction is essential due to the large number of time steps. Since we are primarily focused on detecting changes in wave behavior caused by cracks, forwarding only the strongest signals simplifies the model without sacrificing key information. Waves behave most distinctively when they encounter a crack, which is our highest priority. By downsampling, we effectively reduce model complexity, and this downsample factor of 4 was chosen after extensive evaluations with different max-pooling sizes. This is followed by a series of four convolutional blocks, each progressively increasing the complexity of the feature extraction process.

**Convolutional Blocks**  Each convolutional block is designed to capture different levels of features from the input. The convolutional layers within each block use kernel sizes of (1x1), (3x3), and (5x5), with filters increasing progressively across the blocks. The filter size starts at 16 for the first block and increases by a factor of 2 for the next blocks, ensuring deeper layers extract more complex features.

Convolution Branches:

- 1x1 Convolution Branch: This branch applies 1x1 convolutions to maintain the spatial dimensions while enhancing feature extraction. It allocates a quarter of the total filters to this branch, helping to reduce parameters and ensure a focused feature transformation.

- 3x3 Convolution Branch: The 3x3 convolutional layers in this branch take half of the total filters, designed to capture more local information from the input. This helps in capturing small-scale patterns that may not be picked up by the 1x1 branch.

- 5x5 Convolution Branch: Allocating another quarter of the total filters, this branch focuses on capturing larger-scale features, extracting more global patterns from the input data.

- Pooling Branch: Alongside the convolution branches, a pooling branch performs MaxPooling (3x3) with strides of (1,1), followed by a 1x1 convolution to further process the pooled features. This helps capture features at multiple scales, including lower-resolution patterns.

After the convolution and pooling operations, the outputs from all branches are concatenated, combining the information extracted at various scales. The resulting feature map captures both time and spatial relationships between wave data and cracks. At this stage, the feature map focuses on both time-based wave behavior and spatial relationships in the 9x9 sensor grid, which is critical for detecting and localizing cracks accurately. After each convolutional block, MaxPooling is applied to reduce the spatial dimensions while retaining the most relevant features. Reducing dimensions through MaxPooling is computationally cheaper than doing so through additional convolutional layers. MaxPooling also helps forward only the strongest signals, which is crucial for wave-based crack detection. We chose MaxPooling over average pooling because even after dimension reduction, the strongest signals carry the most valuable information for identifying cracks. A self-attention mechanism is introduced after each pooling step, which allows the model to focus on specific regions of the feature maps, particularly helping the model focus on cracks in the image. This self-attention layer recalculates the importance of different regions and improves feature selection before passing the feature map to the next convolutional block. The attention mechanism processes both the temporal and spatial information, ensuring that the model focuses on the most relevant signals. The feature map from the last attention layer is passed through a Flatten layer, which converts it into a single-dimensional embedding. This embedding is then fed into a series of dense layers. The first dense layer has 128 output units, which are reduced by half in each subsequent layer, ultimately leading to the final dense layer with 4 output units. This design progressively refines the feature representation, with dropout layers included between dense layers to prevent overfitting.

## EVALUATION

### 3.4 DATASETS

Generating numerical data through real-world experiments would be highly expensive and time-consuming. Therefore, to overcome these challenges, we synthetically generate the data using dynamic Lattice Element Method (dLEM) approach(Moreh et al., 2024). This allows us to model and track wave propagation through cracked materials in a controlled, cost-effective manner, while still maintaining the complexity and realism needed for effective crack detection. By leveraging this synthetic data, we can explore various crack patterns and wave interactions that would be difficult to replicate in real-world settings. The dataset contains information about the behavior of seismic waves the waves propagation in a dynamic lattice model as they move through materials with cracks. The lattice is made up of interconnected elements, and as waves travel through these elements they experience changes in force, displacement, and velocity. As the wave propagates, forces cause displacements, which lead to motion through the lattice structure. The displacement changes, represented by the wave motion, are tracked over 2000 time stamps for each sample. There are total 9x9 sensors to collect this information about the wave behaviour. The dataset captures wave propagation in a dynamic lattice model to observe how seismic waves move through materials with cracks. As waves travel through the interconnected elements of the lattice, they experience changes in force, displacement, and velocity. These displacement changes, representing wave motion, are tracked over 2000 time stamps for each sample. The data is collected by a 9x9 grid of sensors, which records the behavior of the waves as they interact with cracks. In this research, bounding key-points are generated from segmentation labels, which are treated as 2D arrays where the presence of a crack is indicated by a label of '1' and its absence by a '0'. The process involves iterating through each segmentation label to identify the positions of the object within the segmentation label. For each segmentation label, the method locates the positions where the crack is present. If no crack is detected, a 'None' value is assigned to represent the absence of an object. When a crack is found, the minimum and maximum coordinates that enclose the crack are calculated. To ensure the bounding key-points provides a slightly wider margin around the crack, a one-pixel margin is added to both the minimum and maximum coordinates. These minimum and maximum coordinates are normalized between 0 and 1 relative to the size of the segmentation label.

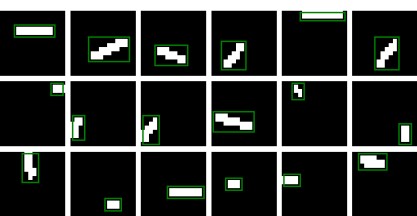

Figure 2: Sample labels for the dataset

## 3.5 METRICS

### 3.5.1 INTERSECTION OVER UNION:

The *Intersection over Union (IoU)* metric is a standard evaluation measure used to assess the accuracy of object detection models. It quantifies the overlap between the predicted bounding box and the ground truth bounding box of an object. IoU is calculated by dividing the area of overlap between the two bounding boxes by the area of their union. The mathematical formulation is:

$$\text{IoU} = \frac{\text{Area of Overlap}}{\text{Area of Union}} = \frac{|A \cap B|}{|A \cup B|}$$

In this equation:

- $A$ represents the predicted bounding box.
- $B$ represents the ground truth bounding box.
- $|A \cap B|$ is the area of overlap between the two boxes.
- $|A \cup B|$ is the total area covered by both boxes combined.

This metric provides a straightforward way to measure how accurately the model predicts the location and size of objects, with higher IoU values indicating better performance.

In this study for evaluating localization quality in object detection, two complementary metrics are used: **Purity** and **Integrity** (Table 2 )(Gao et al., 2022).

**Purity** quantifies how much of the predicted bounding key-points corresponds to the object of interest. It is defined as:

$$\text{Purity} = \frac{\text{Overlap Area}}{\text{Detected Area}}$$

This metric emphasizes the precision of the predicted bounding key-points, focusing on minimizing the inclusion of irrelevant areas.

**Integrity** measures the extent to which the detected bounding key-points covers the ground-truth object. It is calculated as:

$$\text{Integrity} = \frac{\text{Overlap area}}{\text{Ground Truth Area}}$$

Integrity highlights the completeness of the object's coverage, ensuring that the detected bounding key-points encompasses the object without missing significant parts.

By considering both Purity and Integrity, two essential aspects of detection quality are captured: the accuracy of the bounding key-points placement (Purity) and the thoroughness of crack coverage (Integrity). This decomposition allows for separate modeling and prediction of the precision and completeness of Crack localization, simplifying the complex task of directly estimating IoU. By focusing on Purity and Integrity individually, more accurate assessments of localization confidence are achieved, ultimately improving object detection performance. The model achieves an MAE of **0.0731**, indicating that its predictions, on average, deviate by about 0.07 unit from the actual values, demonstrating good accuracy. The MSE of **0.0134** suggests that while the model performs well overall, it may encounter occasional larger errors, though they are not significant. The Huber Loss of **0.0067** further confirms the model's robustness by balancing the effects of small and large errors. Overall, the New Model shows strong predictive performance with minimal errors and resilience to outliers.

Table 1: Crack Size vs IoU, Purity and Integrity

| Crack Size | IoU | Purity | Integrity |
|:---:|:---:|:---:|:---:|
| $> 0$ | 0.511231 | 0.654841 | 0.677959 |
| $>0.0010$ | 0.534543 | 0.685747 | 0.703990 |
| $>0.0020$ | 0.579990 | 0.747775 | 0.745342 |
| $>0.0030$ | 0.608330 | 0.791077 | 0.755843 |
| $>0.0040$ | 0.631770 | 0.829386 | 0.760361 |

### 3.5.2 COMPARISON WITH THE PREVIOUS MODEL

In MicroCrackPointNet, the IoU is calculated by comparing the predicted and ground truth bounding key-points. These boxes represent the regions where cracks are located. The algorithm calculates the intersection of the predicted and actual bounding key-points by finding the overlapping area between them. It then computes the area of the union, which is the combined area of both the predicted and true bounding key-points. Finally, the IoU is calculated as the ratio of the intersection area to the union area. This method focuses on how well the model localizes cracks, with IoU values depending on how accurately the predicted bounding key-points align with the actual cracks.

In contrast, the 1D-Densenet200E model Moreh et al. (2024) calculates IoU by applying a threshold to the predicted crack probability map, converting it into a binary mask. The model then uses a confusion matrix to compute IoU, focusing on pixel-level accuracy in identifying crack regions. True positives, false positives, and false negatives are computed, and the IoU is derived by dividing the true positive area by the sum of the true positives, false positives, and false negatives. This approach is well-suited for crack segmentation tasks, where each pixel's classification matters. As seen in the table 2, 1D-Densenet200E achieves higher IoU values compared to the current study, this stems from the difference in the approach in calculating the IOU, where the dense-net uses a thresholding value to binarizie its output. Moreover, the numerator of the IOU is skewed because of large number True Negatives (No Crack Pixels) in case of 1D-Densenet200E.

Table 2: IOU for various crack sizes for MicroCrackPointNet and 1D densenet

| Crack Size | MicroCrackPointNet | 1D-Densenet200E |
|:---:|:---:|:---:|
| $> 0$ | 0.511231 | 0.6694 |
| $>0.0010$ | 0.534543 | 0.7181 |
| $>0.0020$ | 0.579990 | 0.7631 |
| $>0.0030$ | 0.608330 | 0.7672 |
| $>0.0040$ | 0.631770 | 0.7719 |

### 3.6 A NOVEL APPROACH FOR CRACK DETECTION IN NUMERICAL, NON-VISUAL DATA USING DEEP LEARNING

Crack detection in large structures poses a substantial challenge, especially when dealing with highly imbalanced datasets. Traditional pixel-wise classification methods, which predict each pixel as either "crack" or "no crack," often struggle due to the small area occupied by cracks relative to the overall surface, leading to a significant imbalance in the dataset. This imbalance increases model complexity and demands extensive optimization to achieve accurate results. In this paper, we introduce an innovative approach that addresses these limitations by focusing on numerical, non-visual data. Our method leverages deep learning techniques to overcome the inefficiencies of traditional pixel-level classification, offering a more effective and streamlined solution for crack detection in complex and imbalanced datasets.

### 3.7 OUR KEYPOINT-BASED APPROACH

To address these challenges, we propose an alternative method that detects cracks by identifying four key points, or keypoint, that define the corners of a bounding rectangle around each crack. Rather than making predictions for every pixel, our model predicts only the coordinates of these four points (eight values representing the $x, y$-coordinates). This reduces the model's complexity and alleviates

the issue of class imbalance, as it eliminates the need for pixel-wise decisions. By focusing on the precise localization of cracks through these key-points, the model is optimized for both performance and efficiency, providing a streamlined solution to crack detection.

### 3.8 CRACK DETECTION IN NUMERICAL, NON-VISUAL DATA

A key innovation in our work is the application of this approach to numerical data, where cracks are not visually discernible. In traditional applications such as visual crack detection on concrete or metal surfaces, cracks can be visually identified by humans from image data. However, in our case, the input data consists solely of numerical measurements, which contain no obvious visual patterns. This type of data is impossible for human observers to interpret in terms of crack presence or structure, as there are no visible clues. Here, deep learning demonstrates its power: our model is capable of learning hidden patterns and detecting cracks with precision, despite the absence of visual information. This capability sets our method apart from conventional approaches and highlights the potential of neural networks to operate in domains where visual indicators are not present.

### 3.9 CHALLENGE: OPTIMIZING THE NUMBER OF KEY-POINTS FOR COMPLEX CRACK STRUCTURES IN LARGE-SCALE SURFACES

A key limitation of our current approach is its ability to detect only a single crack per sample using a fixed number of four key-points. This method is effective for simple, rectangular cracks but faces challenges with multiple cracks or more complex geometries. The fixed key-points structure may be insufficient for capturing intricate crack patterns, potentially limiting the model's applicability to diverse real-world scenarios. Despite these limitations, we are confident that the foundational approach we have demonstrated is capable of scaling to handle more complex crack geometries. We anticipate that increasing the number of key-points will enable the model to better represent varied crack shapes and detect multiple cracks within a single structure. Future work will focus on addressing these complexities, optimizing the number of key-points, and refining the model to enhance its accuracy and efficiency for real-world applications.

### 3.10 RESULTS AND IMPLICATIONS

Our experiments show that the keypoint-based approach not only reduces complexity and tackle the class imbalance issue, but also enables accurate crack detection in numerical, non-visual datasets. This represents a significant advance in the field of crack detection, where prior work has been heavily reliant on visual data. The ability of our model to learn numerical patterns and precisely localize cracks opens up new possibilities for applications in domains where visual data is either unavailable or non-informative.

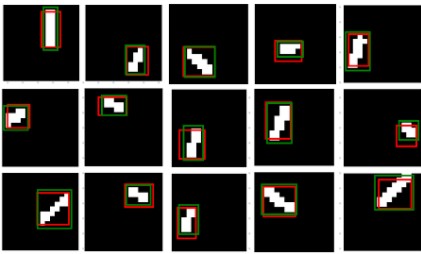

Figure 3: Predicted results where red bounding boxes are predicted and green bounding boxes are ground truth

The results of the proposed models can be categorized based on their performance in crack detection. In Figure 3, MicroCrackPointNet demonstrates good performance, where the detected cracks (green bounding boxes) are sharp and closely align with the ground truth. Figure 4 presents a less favorable outcome, where the model detects only part of the crack, leaving some areas uncovered. This indicates a lack of precision in crack localization. Conversely, Figure 5 illustrates cracks that model could not accurately, highlighting areas for improvement. These instances, particularly involving microcracks where the change in the wave is not significant and easily detectable by model, suggest

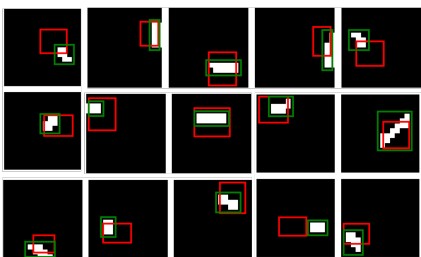

Figure 4: Predicted results where red bounding boxes are predicted and green bounding boxes are ground truth

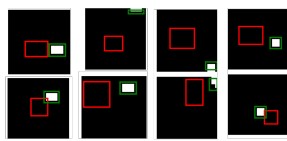

Figure 5: Predicted results where red bounding boxes are predicted and green bounding boxes are ground truth

future research directions aimed at enhancing the detection capability for smaller, less prominent cracks. Overall, the Squeeze and Excite mechanism in MicroCrackPointNet proves advantageous, setting a promising foundation for further advancements in crack detection models.

### 3.11 COMPARATIVE ANALYSIS OF MODELS FOR COMPUTING POWER UTILIZATION

The comparison between the MicroCrackPointNet and the previous model, 1D-DenseNet200E (Table 3), highlights several key improvements in efficiency. The New Model has a more compact architecture, with only 90 layers compared to 444 in the 1D-DenseNet200E. Both models were trained for 200 epochs, but the New Model is much faster, with the first epoch taking 17.03 seconds versus 89.14 seconds for 1D-DenseNet200E. The total training time is also significantly shorter, with the New Model completing training in 2160.53 seconds compared to 15560.56 seconds. The New Model also has fewer total parameters (1,228,760 vs. 1,393,429) and significantly fewer non-trainable parameters (1,544 vs. 17,292), while maintaining a comparable number of trainable parameters. Overall, the New Model is more efficient in terms of both complexity and training time.

Table 3: Comparitive analysis based on parameters and training time

| Attributes | MicroCrackPointNet | 1D-DenseNet200E |
|---|---|---|
| Layers | **90** | **444** |
| Epochs | 200 | 200 |
| Time taken by first Epoch | 17.03 sec | 89.14 sec |
| Total training time | **2160.53 sec** | **15560.56 sec** |
| Total params | 1,228,760 | 1,393,429 |
| Trainable params | 1,227,216 | 1,376,137 |
| Non-trainable params | 1,544 | 17,292 |

## CONCLUSION

In this work, we introduced a novel approach to crack identification in large structures that overcomes the limitations of traditional pixel-wise classification methods. By utilizing key landmarks

to define cracks and applying the method to numerical, non-visual data, we demonstrated that our network can detect hidden patterns in spatio-temporal data that are imperceptible to humans. This approach simplifies the issue of imbalanced datasets by shifting the model's focus from making a decision for every pixel (crack or no crack) to a much easier task—predicting four key points and aligning them as closely as possible with the four ground-truth points, minimizing the distance between them. A major advantage of this method is that it alleviates the computational burden of pixel-wise classification, allowing for more efficient processing. Our relatively simple model achieved an IOU score of 0.511 on a label size of 16x16, a Purity of 0.654841, and an Integrity of 0.677959. These results pave the way for keypoint-based detection methods, demonstrating that crack detection does not necessarily require highly complex models, but can instead focus on more efficient, keypoint-based predictions. Moreover, this technique shows great potential for application in areas where high-dimensional numerical data plays a central role, such as structural monitoring and damage detection, where traditional image-based methods fall short. The ability of our model to focus on key structural points rather than individual pixels enables it to perform robustly in scenarios with sparse or imbalanced data, making it particularly suited for practical applications in resource-constrained environments. Beyond its immediate application to crack detection, this approach opens the door to further research and potential applications in diverse fields such as medical imaging or infrastructure monitoring, where simplified yet effective segmentation models could have a significant impact. Our method demonstrates that by focusing on the core structural elements of a problem—represented as key-points —the complexity of the task can be reduced without sacrificing accuracy. Overall, this keypoint-based approach offers a promising advancement in the field of crack identification, improving both the efficiency and accuracy of segmentation in highly imbalanced datasets while providing a flexible framework for future research in areas where numerical data is paramount.

## FUTURE WORK

In future research, we aim to enhance our approach by refining the number and placement of key points used for crack labeling. Initially, we employed four points to mark the corners of a rectangle, as this was effective for the cracks in our data, which often exhibited rectangular shapes. However, real-world cracks frequently have more complex geometries. To address this, we plan to explore the use of a variable number of key points that can better capture these diverse structures. By increasing or adapting the number of points based on the crack's shape, the model can more accurately represent intricate geometries, leading to more precise predictions. Additionally, we intend to investigate object detection algorithms like YOLO (You Only Look Once) and Faster R-CNN to address the limitation of detecting only one crack per sample. These models could be adapted to recognize multiple cracks simultaneously by outputting several sets of bounding keypoint for each detected crack. This would make the model more robust, allowing it to detect multiple cracks in a single sample, especially when cracks are located in close proximity. We also aim to extend our approach by testing it on higher-resolution datasets. Additionally, while our current dataset consists of samples with only a single crack, we plan to introduce samples containing multiple cracks. By increasing the resolution, we hope to further improve the model's ability to generalize and detect cracks of varying sizes and complexities, which will enhance its robustness and real-world applicability. Another line of exploration is integrating recurrent neural networks (RNNs) alongside convolutional neural networks (CNNs). Our numerical data is three-dimensional, comprising time, $x$, and $y$ coordinates of the surface. Cracks are identified based on how waves propagate across the surface, with sensors capturing wave patterns. When cracks are present, the waves are disrupted, revealing distinct patterns. Given the sequential nature of this data, RNNs could model the temporal dependencies more effectively, enhancing the model's ability to detect cracks based on evolving wave patterns. Furthermore, now that our approach focuses on predicting key points rather than classifying individual pixels, the problem of imbalanced data is less significant. This allows us to explore higher-dimensional datasets, where cracks and structural disruptions may have more complex representations. Expanding our method to handle higher-dimensional data would further validate the model's adaptability and effectiveness across a wider range of applications, improving its generalization to different types of surfaces and crack patterns.

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
