# OpenReview forum: "Deep Learning for Micro-Scale Crack Detection on Imbalanced Datasets Using Key Point Localization"
_ICLR.cc/2025/Conference — Submitted to ICLR 2025_

### Official Review · Reviewer_T85A · 2024-10-31

**Soundness:** 2
**Presentation:** 2
**Contribution:** 2
**Rating:** 3
**Confidence:** 5

**Summary:**

The manuscript presents a method for crack detection using a keypoint-based approach, focusing on the identification of bounding rectangle corners around cracks in numerical, non-visual data.

**Strengths:**

1) The work designed a a keypoint-based method for crack detection. It shows good performance on tiny cracks.
2) The idea in the manuscript is clear and easy to follow.

**Weaknesses:**

- The motivation is not clearly introduced. The difference between the detection micro-cracks and common cracks is not clear. As we know, the tiny cracks captured by high-resolution cameras will also show large width. The problem of detecting tiny crack will generate into that of detecting common cracks. What's the problem of detecting cracks in the visual perception perspective?
- The authors employ Inception-like wide convolutions, but the only model compared is a segmentation method based on DenseNet. Recent advancements in fields such as crack detection and edge detection, including methods like EfficientCrackNet and SegFormer, are notably absent from the comparisons. Including these would provide a more comprehensive evaluation of the proposed model's performance against current state-of-the-art techniques.
- The discussion around the model's performance in cases of closely spaced or overlapping cracks is insufficient. More examples and analyses in this area would help clarify the model's limitations.
- While the comparative analysis highlights improvements in training efficiency, it lacks a discussion on the trade-offs involved in model complexity and performance. An exploration of how reduced layers and parameters might impact detection accuracy would be valuable.
- Figures 3-6 depicting results could be made more informative. While they present predicted versus ground truth, additional context or examples of challenging cases would provide a more comprehensive view of model performance.
- The manuscript would benefit from a more polished writing style, as some sections are verbose and could be more concise. Specifically, in Section 3.0.1, the description of wide convolutional networks should be simplified to focus on the core benefits without repetitive elaboration. Additionally, in Section 3.2, the authors only introduce three different loss functions without clearly specifying which one is utilized for model optimization during training. This omission should be addressed to provide clarity on the model's training process.

**Questions:**

What is the motivation of detecting micro-cracks while the problem can be generated into common crack detection under high resolution imaging?
Why not compare the proposed method with recent advanced ones like EfficientCrackNet or SegFormer?
What is the model's performance on crack detection in cases of closely spaced or overlapping cracks?

---

### Official Review · Reviewer_JqDd · 2024-10-31

**Soundness:** 3
**Presentation:** 2
**Contribution:** 1
**Rating:** 3
**Confidence:** 5

**Summary:**

This work proposes a network for detecting crack bounding boxes from seismic wave data. The network is trained and tested using synthetic data generated by the authors. The experimental results demonstrate that the proposed network can effectively detect cracks with reasonable performance across various crack sizes.

**Strengths:**

I can not see significant strengths from this work.

**Weaknesses:**

1. Detection or segmentation internal crack have been widely studied, especially using waveform data, such as seismic and GPR data. To name some, [1] Arbitrary-oriented tunnel lining defects detection from GPR images using deep CNN, [2] Defect segmentation: Mapping tunnel lining internal defects with ground penetrating radar data using a convolutional neural network, [3] GPRI2Net: A deep-neural-network-based ground penetrating radar data inversion and object identification framework for consecutive and long survey lines. Compared with these works, the novelty and insight in this paper are limited.
2. Currently, most crack detection studies utilize real field data from practical projects to ensure generalization. However, this work relies solely on synthetic data.
3. The experiments are insufficient and lack an ablation study.

**Questions:**

As stated in the work “SeisInvNet,” there is a complex relationship between seismic waves and the model, as they originate from two distinct domains: one being spatial and the other spatio-temporal. Consequently, directly applying CNNs for mapping between these domains may not be appropriate. Have you investigated whether CNNs are suitable for addressing your task?

---

### Official Review · Reviewer_gXNj · 2024-11-01

**Soundness:** 1
**Presentation:** 1
**Contribution:** 1
**Rating:** 1
**Confidence:** 5

**Summary:**

This paper presents a CNN-based approach to detect internal cracks by using wave signals. The authors formulate the crack detection problem as a task to predict a bounding box which tightly contains the internal cracks in a manner similar to object detection. A CNN is composed of inception-like multi-branch modules to effectively encode input signals in a framework of bounding-box regression. In the experiments on crack detection using synthetic wave signal datasets, the method exhibits favorable performance.

**Strengths:**

+ The authors tested a smaller network than 1D-DenseNet [Moreh+24] on a signal-based crack detection task.

**Weaknesses:**

- Unfortunately, it is quite hard to find sufficient technical novelty in this work. All the components presented in this paper are directly derived from prior works; Inception modules [Szegedy+14] are used to construct the network which is trained based on a standard loss functions such as MSE, MAE and Huber losses, and then it is tested on the dataset synthesized by [Moreh+24].

- This paper lacks clear advantage to formulate the crack detection as bounding-box regression over a standard semantic segmentation approach [Moreh+24]. Generally speaking, bounding boxes would miss capturing detailed structure/shape of cracks; for example, length, width and (dis-)connectivity of cracks are important for structural health monitoring, which bounding boxes are unaware of. As to performance, the experimental results do not clearly show superiority of the bounding box approach.

- In this community, *width* of networks is usually indicated by the number of filters [Zagoruyko+17]. Although the authors claim to propose  *wide* network, it is just composed of inception modules without paying much attention to the number of filters. Thus, the authors' claim is misleading.

- Statements in Sec.1 are rather exaggerated. The crack-detection task that this paper tackles stems from [Moreh+24].

- This paper is poorly presented.
  - The structure, e.g., dimensions, of input signals is unclear in Secs.1~3. Thus, it is less understandable why 2-D CNN is applied to this task. Fig.1 that shows an input as 1-D signal is quite misleading since the input is actually shaped as 3-D tensor of H x W x Time.

  - There are some redundant descriptions; Sec.3.3 is heavily overlapped with Sec.3.1, and it does not mention the training procedure. Sec.3.4 (*Datasets*) is also described in a redundant manner.

### Minor comments
- Section numbers in Sec.*Evaluation* are wrong.

**Questions:**

See the above-mentioned weak points.

---

### Official Review · Reviewer_c5Z4 · 2024-11-03

**Soundness:** 3
**Presentation:** 2
**Contribution:** 2
**Rating:** 3
**Confidence:** 3

**Summary:**

This paper proposes a novel approach for micro-scale crack detection using deep learning (DL) methods that employ key point localization to mitigate class imbalance. The proposed model, named MicroCrackPointNet, is a wide convolutional network designed to predict the coordinates of four key points that define a bounding region around cracks. The paper utilizes non-visual numerical wave propagation data, making it suitable for detecting cracks not visible to the human eye. With an IoU score of 0.511 and Purity and Integrity scores of 0.654 and 0.678, respectively, the model offers a promising, computationally efficient solution compared to traditional segmentation models.

**Strengths:**

1. **Key-Point Localization Approach**: By predicting key points instead of conducting pixel-wise segmentation, the model efficiently addresses the class imbalance problem and reduces computational complexity.

2. **Application on Non-Visual Data**: The model’s application to non-visual data is valuable for industrial applications where cracks may not be visually detectable, demonstrating DL's potential beyond typical computer vision tasks.

3. **Computational Efficiency**: The model’s lightweight architecture and key-point-based approach provide a faster training time and lower computational load compared to prior dense segmentation models, as evidenced by reduced parameter counts and training time.

**Weaknesses:**

1. **Baselines and Quantitative Comparisons**: While the paper claims efficiency gains, direct quantitative comparisons with established object detection or segmentation models beyond 1D-DenseNet200E are lacking. This hinders validation of the model’s novelty and its performance relative to state-of-the-art approaches.

2. **Evaluation Metrics**: The model primarily reports IoU, Purity, and Integrity but lacks more nuanced metrics, such as precision and recall, which would provide a clearer picture of the model’s handling of false positives and false negatives.

3. **Generalizability to Complex Crack Patterns**: The model only supports single, rectangular cracks, and struggles with complex or multiple cracks, limiting its applicability to real-world scenarios where cracks often exhibit diverse geometries.

**Questions:**

1. **Scalability to Realistic Crack Patterns and Scenarios**: Given that the model is currently limited to detecting single, rectangular cracks, how does the proposed approach intend to handle diverse crack shapes or multiple overlapping cracks?

2. **Quantitative Validation Against State-of-the-Art Models**: The paper lacks robust baseline comparisons with well-known object detection models such as YOLO or Faster R-CNN. How does the model perform relative to these benchmarks, particularly on speed and detection quality?

3. **Evaluation Metrics**: Why did the authors not report additional performance measures, such as F1 score or area under the ROC curve (AUC), to better assess the model’s accuracy, especially given the challenges with imbalanced datasets? Would these metrics offer a more comprehensive view of model performance?

4. **Purity and Integrity Metrics**: The Purity and Integrity metrics lack adequate justification for their advantage over IoU. What practical benefits do they offer in this context, and how do they impact model evaluation outcomes differently than IoU?

---

### Official Review · Reviewer_kbsT · 2024-11-03

**Soundness:** 2
**Presentation:** 2
**Contribution:** 2
**Rating:** 5
**Confidence:** 4

**Summary:**

The proposed model identifies and locates hidden cracks by localizing the coordinates of four key points of the bounding region of the crack. They generate synthetic data using dynamic Lattice Element Method (dLEM) approach to model and track wave propagation through cracked materials in a controlled, cost-effective manner, while maintaining the complexity for effective crack detection.

**Strengths:**

The proposed method reduces computational complexity enhancing the efficiency of the crack detection by focusing on predicting four key points rather than analysing every pixel by making use of the seismic waves rather than visual cues to detect cracks.

The proposed model has fewer layers, parameters, and shorter training times for detecting single, rectangular cracks.

The model’s architecture and training process are suitable for detecting single, rectangular cracks, providing targeted performance improvements for this particular crack type.

**Weaknesses:**

The use of key point coordinates to localise cracks can lead to inaccuracies, specially, if the key points are slightly off. This could be particularly problematic for smaller cracks, where minor deviations in key-point placement could significantly impact the model's accuracy.

Even though the paper addresses data imbalance by using key-point detection, significant risk of bias remains. As an example, in real-world scenarios, where crack samples are often sparse and data distributions differ from simulated conditions, the model’s reliance on simulated data may cause it to disproportionately predict non-crack regions.

The model’s current design only allows for the detection of a single, rectangular crack per sample, which may limit its practical applicability. However, many real-world scenarios often involve multiple or irregularly shaped cracks.

**Questions:**

A similar study, “Deep neural networks for crack detection inside structures” by Moreh, F., Lyu, H., Rizvi, Z.H., and Wuttke, F. (2024) published in Scientific Reports, also employs numerical wave data generated through dLEM simulations for crack detection. Could you clarify the distinctions between their work and yours, particularly in light of your statement that this is the first study to apply this approach to crack detection using numerical data? In what ways does your methodology differ from theirs?

Do you consider the use of attention layer is sufficient for noise mitigation? Can you provide further insights into the model's sensitivity to mitigate noise and any additional strategies to ensure robustness in noisy environments?

---

### Meta-Review · Area_Chair_jggo · 2024-12-18

**Metareview:**

The paper introduces a deep learning approach for detecting micro-scale cracks in structural materials by predicting the coordinates of four key points that define a bounding box around the crack. This method aims to reduce computational complexity, address the challenge of imbalanced datasets, and improve crack detection for non-visual data derived from seismic wave propagation. While the concept of using keypoint localisation for crack detection is novel for this application, the reviewers raised significant concerns related to technical novelty, experimental rigour, generalisability, and clarity, which ultimately led to the paper's rejection.

The approach was criticised for its reliance on existing methods, particularly the use of Inception modules and standard loss functions like MSE, MAE, and Huber losses. Many reviewers argued that the core components of the model were directly drawn from prior works, with little technical innovation. The choice to frame crack detection as a bounding box regression task rather than using semantic segmentation was seen as a suboptimal approach. Bounding boxes are considered less effective for representing cracks since they fail to capture finer details like crack width, length, and connectivity.
Claims of introducing a "wide convolutional network" were viewed as misleading since the width of the network was not explicitly controlled (e.g, by increasing filter counts), but rather achieved through Inception-style multi-branch layers.
The motivation for targeting "micro-crack detection" was questioned, as it was argued that micro-cracks could be addressed with high-resolution imaging and general crack detection methods.

The experimental validation was seen as inadequate, with reviewers calling for a more extensive evaluation of the model. Key missing elements included:
Ablation studies to isolate the contribution of keypoint localisation relative to other model components.
Comparisons with state-of-the-art crack detection models, such as EfficientCrackNet or SegFormer, as well as traditional object detection models like YOLO or Faster R-CNN.
The evaluation relied heavily on synthetic data generated through seismic wave simulations, with no validation on real-world datasets. Reviewers argued that synthetic data might not fully reflect the complexities of real-world crack patterns, limiting the model's generalization potential.
The paper reported performance using IoU, Purity, and Integrity metrics, but reviewers questioned why more common metrics like precision, recall, F1-score, and AUC were not included. The use of Purity and Integrity was criticised for lacking sufficient justification as to why they were more informative than standard metrics.

Lastly, the paper's structure was seen as redundant and unclear. Multiple reviewers noted overlaps between sections, particularly between Sections 3.1 and 3.3, which repeated technical descriptions. This redundancy made it difficult to understand the unique contributions of different parts of the model.

I would encourage the reviewers to improve their paper and resubmit to another venue.

**Additional Comments On Reviewer Discussion:**

The authors decided to not submit a rebuttal. Therefore the discussion has been limited.
Nevertheless, the reviewers have agreed that the paper is not suitable for publication in its current shape.

---

### Decision · Program_Chairs · 2025-01-22

Reject